# Structure–Functional Examination of Novel Ribonucleoside Hydrolase C (RihC) from *Limosilactobacillus reuteri* LR1

**DOI:** 10.3390/ijms25010538

**Published:** 2023-12-30

**Authors:** Leonid A. Shaposhnikov, Natalia Yu. Chikurova, Denis L. Atroshenko, Svyatoslav S. Savin, Sergei Yu. Kleymenov, Alla V. Chernobrovkina, Evgenii V. Pometun, Mikhail E. Minyaev, Ilya O. Matyuta, Dmitry M. Hushpulian, Konstantin M. Boyko, Vladimir I. Tishkov, Anastasia A. Pometun

**Affiliations:** 1Bach Institute of Biochemistry, Federal Research Centre “Fundamentals of Biotechnology” of the Russian Academy of Sciences, Leninsky Avenue, 33/2, Moscow 119071, Russia; chikurova.nu@yandex.ru (N.Y.C.); atrdenis@gmail.com (D.L.A.); savinslava@gmail.com (S.S.S.); s.yu.kleymenov@gmail.com (S.Y.K.); i.matyuta@fbras.ru (I.O.M.); hushpulian@gmail.com (D.M.H.); kmb@inbi.ras.ru (K.M.B.); vitishkov@gmail.com (V.I.T.); 2Department of Chemistry, Lomonosov Moscow State University, Leninskie Gory, 1–3, Moscow 119991, Russia; chernobrovkina@analyt.chem.msu.ru; 3Institute of Medicine, Peoples’ Friendship University of Russia Named after Patrice Lumumba, Miklouho-Maklaya, 8, Moscow 117198, Russia; 4Koltzov Institute of Developmental Biology of Russian Academy of Sciences, Vavilova, 26, Moscow 119334, Russia; 5Department of Analytical, Physical and Colloidal Chemistry, A.P. Nelyubin Institute of Pharmacy, Sechenov First Moscow State Medical University, Trubetskaya St., 8, Building 2, Moscow 119048, Russia; epometun@gmail.com; 6N. D. Zelinsky Institute of Organic Chemistry Russian Academy of Sciences, Leninsky Avenue, 47, Moscow 119991, Russia; mminyaev@ioc.ac.ru; 7Faculty of Biology and Biotechnology, National Research University Higher School of Economics, Profsoyuznaya St., 33, Building 4, Moscow 117418, Russia

**Keywords:** ribonucleoside hydrolase C, lactobacilli, nosocomial infections, enzymatic activity, crystal structure, model structure studies

## Abstract

Ribonucleoside hydrolase C (RihC, EC 3.2.2.1, 3.2.2.2, 3.2.2.3, 3.2.2.7, 3.2.2.8) belongs to the family of ribonucleoside hydrolases Rih and catalyzes the cleavage of ribonucleosides to nitrogenous bases and ribose. RihC is one of the enzymes that are synthesized by lactobacilli in response to the presence of *Klebsiella*. To characterize this protein from *Limosilactobacillus reuteri* LR1, we cloned and expressed it. The activity of the enzyme was studied towards a wide range of substrates, including ribonucleosides, deoxyribonucleosides as well as an arabinoside. It was shown that the enzyme is active only with ribonucleosides and arabinoside, with the best substrate being uridine. The thermal stability of this enzyme was studied, and its crystal structure was obtained, which demonstrated the tetrameric architecture of the enzyme and allowed to shed light on a correlation between its structure and enzymatic activity. Comprehensive comparisons of all known RihC structures, both existing crystal structures and computed model structures from various species, were made, allowing for the identification of structural motifs important for enzyme functioning.

## 1. Introduction

Nosocomial infections are currently a pressing and serious problem. One of the causative agents of these infections is bacteria of the genus *Klebsiella*, which cause pneumonia, sepsis, inflammation of the urinary system, and liver and kidney problems, and their resistance to antibiotics is increasing every year [1]. It is known that when treated with antibacterial agents, these bacteria go into a biofilm state, in which their resistance significantly increases [2]. The problem of *Klebsiella* biofilms is particularly acute in the field of joint transplantation. On average, every 25–30th operation is accompanied by sepsis caused by *Staphylococcus* or *Klebsiella*, which, in this case, exist in the body precisely in the form of biofilms [3]. Therefore, the search for ways to combat such biofilms is relevant at the moment.

When carrying out the co-cultivation of bacteria of the genus *Lactobacillus* and bacteria of the genus *Klebsiella*, it turned out that there are two strains of lactobacilli, *Limosilactobacillus reuteri* LR1 and *Lacticaseibacillus rhamnosus* F, which exhibit an antagonistic effect on *Klebsiella* [4]. These strains are particularly interesting because they are part of the human intestinal microbiota, which means that the proteins they synthesize are highly unlikely to trigger an immune response. Proteomic analysis of both free *L. reuteri* and *L. rhamnosus* and their co-cultivation with bacteria of the genus *Klebsiella* was carried out [4], and it turned out that in response to the action of the pathogen, lactobacilli start synthesizing several proteins in high quantities that help suppress the growth of *Klebsiella*. Analysis using tandem MALDI/TOF/TOF spectrometry showed that among these proteins, there are hydrolases that contribute to the destruction of peptide cross-links in *Klebsiella* peptidoglycan or the carbohydrate backbone of this peptidoglycan, proteins that hydrolyze nucleic acids, as well as metabolic proteins that are synthesized only in response to the presence of *Klebsiella*. One such nucleoside-degrading enzyme is ribonucleoside hydrolase C (RihC) from *L. reuteri*.

RihC catalyzes the cleavage of ribonucleosides to nitrogenous bases and ribose. This enzyme belongs to the family of ribonucleoside hydrolases Rih, which includes several enzymes: IG–NH (inosine–guanosine-preferring nucleoside hydrolase), IAG–NH (inosine–adenosine–guanosine-preferring nucleoside hydrolase), RihA, RihB and RihC [5]. IG–NH and IAG–NH are purine-specific hydrolases, RihA and RihB are pyrimidine-specific hydrolases (also called CU–NH or cytidine–uridine-preferring nucleoside hydrolase), while RihC catalyzes reactions with both pyrimidines and purines [5,6,7]. The physiological role of this enzyme in lactobacilli and why it is synthesized specifically in response to the presence of pathogens is not completely clear.

In this work, the RihC enzyme from *L. reuteri* was cloned, and a comprehensive structural and functional characterization was carried out. We have obtained two forms of this enzyme with different His-tag positions, studied the kinetics of these forms and the thermal stability of the enzyme, obtained a crystal structure, and shed light on the relationship between the structure and the catalytic properties of this enzyme.

## 2. Results and Discussion

### 2.1. Selecting a Proper His-Tag Position

To improve protein purification, a six-histidine tag (His-tag) was added to the RihC construct in two alternatives—to the N– or C–terminus.

Structural modeling was carried out to verify a proper His-tag position using a free version of the AlphaFold2 tool based on Google Collab (ColabFold v1.5.3) [8,9]. Models were built with the addition of a His-tag at the N– or C–terminus of the enzyme to explore whether the His-tag can affect the properties of the enzyme or not and whether it is available for binding to the metal chelate chromatography column.

To date, plenty of crystal structures are known for enzymes from the Rih family, incl. eight structures for RihC from various sources (Table 1), and while these structures are known to be homodimers or homotetramers, there is no strict correlation between the degree of oligomerization and the source of the enzyme.

In this regard, the modeling of LreRihC was carried out with the addition of a His-tag at the N- or C-terminus (hereinafter LreRihC_HisN and LreRihC_HisC, respectively) in the form of a homodimer and homotetramer. The modeling results are presented in Figure 1.

From this figure, it is clear that the tags at the N- and C-terminus of the enzyme are directed outward from the protein globule and should not seriously affect the intact enzyme structure and/or its catalytic properties. So, it was decided to obtain both variants of the LreRihC enzyme (with His-tag at the N- and C-terminus) and compare their properties.

### 2.2. Obtaining the LreRihC Enzymes with His-Tag

It was found that both enzymes with His-tag are expressed in *E. coli* cells in a soluble form, and the yield was approximately 180 mg per liter of nutrient medium for LreRihC_HisN and approximately 60 mg per liter of nutrient medium for LreRihC_HisC, i.e., the position of the His-tag affected the expression level of this protein; LreRihC_HisN was produced approximately three times greater than LreRihC_HisC. We assume that the folding of LreRihC_HisN after synthesis in the cell occurs more efficiently. Thus, the C-terminal His-tag is not optimal for enzyme production.

The degree of purity of the enzyme was confirmed using SDS–PAGE (Appendix A). Tandem MALDI–TOF mass spectroscopy confirmed the sequence of both enzyme forms. The percentage agreement of LreRihC_HisN and LreRihC_HisC after MALDI analysis with the theoretical sequence were both 98% (Appendix A, respectively). Both enzymes were obtained in a water-soluble, highly purified state with a degree of purity of at least 95%. The difference in expression levels for these two forms of the enzyme can also be seen in this figure. The theoretical mass of the LreRihC monomer with His-tag added at either end is 33.38 kDa, which is also consistent with the SDS–PAGE results for both forms of the protein.

### 2.3. Determination of the Oligomeric Composition of the Enzyme

The oligomeric composition of RihC was determined using analytical gel filtration. The calibration curve with LreRihC_HisN plotted on it is presented in Figure 2. For LreRihC_HisC, identical results were obtained during this work.

The theoretical monomer mass of LreRihC with His-tag at either terminus calculated using the amino acid sequence is 33.38 kDa. The molecular mass of samples LreRihC_HisN and LreRihC_HisC obtained using this gel filtration was 121 and 123 kDa, respectively, which corresponds to a tetramer. In general, there is no strict relationship between the source of nucleoside hydrolase and its oligomeric state in solution [7,10,11]. In the case of hydrolase LreRihC, it turned out that it exists in the form of a tetramer, while the position of the His-tag does not affect its oligomeric state.

### 2.4. Kinetic Parameters of LreRihC

Enzyme activity measurements were carried out according to the approach described in the corresponding section of the experimental part and more thoroughly discussed in [12]. Four main ribonucleosides were chosen as substrates: uridine, cytidine, guanosine, and adenosine, as well as inosine, xanthosine, thymidine, 2′-deoxyribouridine, 5-methyluridine (also known as ribothymidine) and vidarabine. For each of the substrates, the enzymatic activity was measured; for some substrates, the dependence of the rate of the enzymatic reaction on the concentration of the substrate was obtained, and the kinetic parameters were calculated. Firstly, the enzyme’s kinetics with uridine (for both forms) were studied several times with a freshly purified enzyme batch to see if the kinetics converged within the margin of error. Both K_M_ and *k_cat_* values for both forms were the same within the error. Secondly, for each of the substrates studied in terms of kinetics, we purified a new batch of both forms of the enzyme and repeated uridine kinetic studies each time to ascertain that the kinetic parameters obtained were correct while also repeating studies for that substrate once again to be sure that the kinetic values are correct. In total, for both forms of the enzyme for each substrate, the kinetics were studied twice (each substrate with a different batch of freshly purified enzymes), except for uridine, for which the kinetics were studied twice initially and then once each time parallel with the new substrate. All the kinetic parameters for each substrate were the same within the error margin.

The kinetic parameters are presented in Table 2. This table also shows the kinetic parameters of other ribonucleoside hydrolases, C. Appendix A shows the dependence of the enzymatic reaction rate on the substrate concentration in the concentration range of 0.5–5K_M_, which was used to calculate the kinetic parameters for the LreRihC_HisN (Appendix A) and LreRihC_HisC (Appendix A) enzymes. For uridine and cytidine, it was possible to measure enzymatic activity in a concentration range of up to 100 mM for both forms of the enzyme; for adenosine, this was up to 30 mM (due to the poor solubility of this substrate when trying to obtain higher concentrations under conditions optimal for the enzyme to work); for inosine, this was up to 20 mM; and for xanthosine, this was up to 10 mM. Guanosine has low solubility under the reaction conditions; therefore, for this ribonucleoside, it was only possible to measure enzyme activity at a maximum substrate concentration of 2 mM. For vidarabine and ribothymidine, it was also decided to measure only the enzymatic activity with 2 mM of each substrate to compare to the activity with other substrates. The cleavage reactions of thymidine and 2′-deoxyuridine were not observed at any concentration of these substrates, from which it can be concluded that the LreRihC enzyme indeed catalyzes cleavage reactions of ribonucleosides. Moreover, since the enzyme showed activity with both purine and pyrimidine ribonucleosides, this enzyme truly belongs to the RihC class, i.e., nonspecific nucleoside hydrolases.

For all studied substrates for which the kinetic parameters of the enzymatic reaction were obtained, classical Michaelis–Menten dependence is observed for both forms of the LreRihC enzyme at low substrate concentrations, as seen in Appendix A. Moreover, this should not create problems with the use of this enzyme since it is known that the physiological concentration of nucleosides and nitrogenous bases in humans in blood plasma and other extracellular fluids is 0.4–6 μM [13]. Intracellular concentrations are usually slightly higher. Unfortunately, there are no exact data on the physiological concentrations of nucleosides in bacteria, but we assume that they should not greatly exceed the concentrations of these substances in humans.

**Table 2 ijms-25-00538-t002:** Kinetic parameters of RihC enzymes with different substrates. Parameters for the enzymes obtained in this work are in bold.

Enzyme	LreRihC_HisN	LreRihC_HisC	EcoRihC ^b^	CfaRihC	SenRihC	LmaRihC
***k_cat_^uridine^*, s^−1^**	**167 ± 6**	**134 ± 6 ^a^**	10.85 ± 0.23	143	46 ± 3 ^c^	32 ± 6
**K_M_^uridine^, µM**	**320 ± 40**	**320 ± 40 ^a^**	408 ± 184	1220 ± 40	1060 ± 100 ^c^	234 ± 112
***k_cat_^cytidine^*, s^−1^**	**112 ± 4**	**58 ± 6**	1.12 ± 0.53	20	7.8 ± 1.3 ^c^	0.36 ± 0.05
**K_M_^cytidine^, µM**	**680 ± 80**	**620 ± 60**	682 ± 298	4700 ± 500	9200 ± 1200 ^c^	422 ± 175
***k_cat_^inosine^*, s^−1^**	**30 ± 5**	**18 ± 6**	4.31 ± 0.22	32	9.0 ± 0.15 ^c^ 8.1 ± 0.14 ^d^	119 ± 34
**K_M_^inosine^, µM**	**2500 ± 600**	**2600 ± 600**	422 ± 225	380 ± 30	650 ± 60 ^c^ 1280 ± 130 ^d^	445 ± 209
***k_cat_^xanthosine^*, s^−1^**	**57 ± 8**	**40 ± 6**	6.30 ± 0.05	ND	62 ± 8 ^c^ 26.3 ± 0.3 ^d^	ND
**K_M_^xanthosine^, µM**	**1200 ± 200**	**1300 ± 200**	454 ± 165	ND	5900 ± 1000 ^c^ 790 ± 50 ^d^	ND
***k_cat_^adenosine^*, s^−1^**	**118 ± 4**	**65 ± 7**	1.15 ± 0.47	4.3	2.06 ± 0.07 ^c^	0.57 ± 0.04
**K_M_^adenosine^, µM**	**420 ± 50**	**480 ± 90**	416 ± 249	460 ± 30	160 ± 20 ^c^	185 ± 46
***k_cat_^guanosine^*, s^−1^**	**ND**	**ND**	ND	2	ND	0.59 ± 0.03
**K_M_^guanosine^, µM**	**ND**	**ND**	ND	420 ± 10	ND	140 ± 23
**Source**	**This work**	**This work**	[7]	[10,14]	[15]	[14]

ND—no data; ^a^—data obtained in [12]. ^b^—rate constants for this enzyme are given for one subunit by the authors; ^c^—data for pH 7.2; ^d^—data for pH 6.0.

From the table, it is clear that the position of the His-tag does not affect the Michaelis constants of the enzyme LreRihC with all substrates, but it does affect the value of the reaction rate constant *k_cat_*, and for each of the substrates, it is observed that *k_cat_* of LreRihC_HisN is higher than *k_cat_* of LreRihC_HisC. The reaction of uridine cleavage for both forms of this enzyme has the lowest K_M_ and the highest *k_cat_*, which indicates the greatest efficiency of the enzyme with this substrate. It was not possible to measure the kinetic parameters for the reaction with guanosine due to the low solubility of this substrate under the operating conditions of the enzyme. When comparing the kinetic parameters of the LreRihC enzymes obtained in this work with enzymes from other sources, it can be noted that the Michaelis constants are comparable to those of other enzymes, although for LreRihC, the least preferrable is inosine; however, the catalytic constants, even for the least preferred LreRihC substrate, are quite high compared to all other RihC enzymes (comparable to *k_cat_* for more preferred substrates for other RihC enzymes). Overall, our enzyme seems to have high turnover rates for all the substrates, which may possibly play a part in the enzyme’s role when it is produced by *L. reuteri* in response to the presence of *Klebsiella*.

To understand the relative efficiency of guanosine cleavage by this enzyme, a comparison of the activity of the enzyme with different substrates at the same concentrations was made (a concentration of 2 mM was chosen as it was the maximum achievable concentration of guanosine in the working conditions). In addition, the activity of the enzyme was studied in relation to other nucleosides: two 2′–deoxyribonucleosides (thymidine and 2′–deoxyuridine), one ribonucleoside (5-methyluridine) and one arabinoside (vidarabine, an analog of adenosine). The results are presented in Figure 3A.

From the presented data, it is clear that with all basic ribonucleosides, both forms of the enzyme exhibit catalytic activity, with the greatest activity observed for uridine and the least for guanosine. The reaction does not occur with 2′–deoxyribonucleosides, which confirms the important role of the 2′–OH group of the ribonucleoside in catalysis since this group is coordinated by the calcium ion in the active site of the enzyme, thus correctly orienting the substrate for catalysis. Moreover, with vidarabine (an analog of adenosine with arabinose instead of ribose), the 2′–OH group of which is located differently in space than that of adenosine, enzymatic activity is still observed, although it is significantly lower than the activity with adenosine. This means that the 2′–OH position of the sugar is also important for catalysis. From the same figure, it can be seen that the activity of LreRihC_HisC is lower than the activity of LreRihC_HisN for each substrate studied, which is directly related to the kinetic properties of these two forms and confirms the negative effect of C-terminus His-tag on catalytic properties of LreRihC.

To confirm the role of calcium ion in catalysis by this enzyme, the enzymatic activity of uridine cleavage was measured in the presence of 10 mM EDTA, as well as 5, 10 or 50 mM CaCl_2_. These substances were added to enzyme samples, which were then incubated for 24 h at 4 °C, and an enzymatic reaction was performed with 20 mM of uridine. The obtained data are presented in Figure 3B.

The experiment showed that the activity of both enzymes in the presence of EDTA was almost 20 times lower. The addition of calcium ions almost did not change the enzymatic activity for samples with 5 mM and 10 mM CaCl_2_, and for the sample with 50 mM CaCl_2_, the activity increased by 20%. Upon the subsequent addition of calcium ions to the enzyme sample containing EDTA (concentration of CaCl_2_ was 50 mM in the final solution) and incubation, the activity returned to almost the previous level. From this, we can conclude that calcium is indeed important for the catalysis of RihC. EDTA coordinates and pulls calcium ions away from the active site of the enzyme, causing the substrate to not be properly oriented for catalysis and activity to drop significantly. In this case, activity is restored when calcium ions are added to such an enzyme solution. The addition of additional calcium ions has virtually no effect on activity, and the slight increase can be explained by the fact that during protein synthesis in the cells, there may have been such an amount of available calcium ions that a small number of protein globules folded without it. The addition of a calcium solution also allows these certain protein molecules to catalyze reactions, thereby increasing activity.

Since LreRihC_HisN had the best kinetic properties and better expression results, further studies were carried out specifically for this enzyme.

### 2.5. Temperature Stability of LreRihC_HisN

Differential scanning calorimetry (DSC) was used to study the temperature stability of LreRihC. During the experiment, the temperature of the sample cell and the comparison cell was linearly increased, and the change in heat capacity was monitored. Since protein denaturation is a phase transition, the experimental curve shows an increase in the change in heat capacity of the sample with a peak at the temperature at which the maximum rate of denaturation occurs and which is used as a characteristic of the protein’s thermal stability (the so-called phase transition temperature, T_m_). DSC data for LreRihC_HisN are presented in Figure 4A.

The figure shows that T_m_ for this enzyme is 59.5 °C. This DSC curve is fitted by a single calorimetric domain. The protein is irreversibly inactivated.

The kinetics of the thermal inactivation of LreRihC_HisN were also studied at different temperatures. The temperatures chosen were 45, 50, 55 and 60 °C based on DSC data. The dependences of residual enzyme activity on incubation time are presented in the semi-logarithmic coordinates in Figure 4B. All values of residual activity are averaged over three experiments for each point on the graph. To study the activity, 30 mM uridine was used. Enzyme samples were incubated in 0.1 M NaPB pH 7.0.

At 60 °C, enzyme activity drops by two times within 5 min of incubation. In general, all observed dependencies remained linear at different enzyme concentrations, which indicates that the thermal inactivation process is monomolecular. The rate constants of this process for each temperature are presented in the figure. From the data obtained, using the equations of the transition state theory (TST), the activation parameters ΔH^≠^ and ΔS^≠^, as well as the T_20_ parameter (the temperature at which the enzyme activity drops by two times in 20 min), were found. ΔH^≠^ was calculated to be 260 ± 30 kJ mol^−1^, ΔS^≠^ was 490 ± 90 J mol^−1^ K^−1^ and T_20_ was 53.7 °C.

In general, the kinetics of thermal inactivation are in good agreement with the DSC results. At 45 °C, the protein is still mostly stable and has a small thermal inactivation rate constant, which corresponds to the very beginning of the DSC peak; at 50 °C, the inactivation constant increases, but not by a significant amount, and DSC shows that this temperature still corresponds to the beginning of the peak, and at 55 °C and 60 °C, the inactivation constants increase sharply, which corresponds to the maximum of the DSC peak.

### 2.6. Structural Studies of LreRihC

Since structural studies were carried out only for LreRihC_HisN, further in this section, we use the abbreviated name for this enzyme—LreRihC. The crystal structure of LreRihC was determined at a 1.9 Å resolution. There were four almost identical protein subunits (RMSD between subunits do not exceed 0.13 Å) in the asymmetric unit of the crystal, which belongs to the P2_1_ space group. Contact analysis revealed that the protein is a tetramer in a crystal (Figure 5A), which is in accordance with solution studies. A structural comparison of the LreRihC with known structures of RihC from different organisms (Table 1, Figure 5) showed that the assembly is similar to protozoan LmaRihC, which is also tetramer (Figure 5B), while bacterial GvaRihC and plant ZmaRihC are dimers (Figure 5C,D). Despite that, the LreRihC tetramer is organized differently compared to that from LmaRihC, which means different intersubunit interfaces (Figure 5A,B). Table 3 presents all shortened names that are used hereinafter in text for RihCs from different organisms.

The subunit comparison revealed that all the structures have similar folds. However, LreRihC is mostly similar to the protozoan LmaRihC, LbrRihC, and CfaRihC with the least similarity to bacterial GvaRihC and BanRihC (Table 1), and there are three polypeptide regions, which result in high RMSD. Region 1 (residues 272–283 in LreRihC) in all structures is a flexible loop with different conformations (Figure 5E,F). Region 2 (residues 226–234) is unstructured in LreRihC and BanRihC, in contrast to other RihCs (Table 1), where this region is an α-helix. Region 3 (residues 77–99) in LreRihC and LmaRihC has unstructured conformation (Figure 5E), while in GvaRihC and ZmaRihC, this region contains one α-helix (Figure 5F,G).

The LreRihC subunit maintains the typical α/β fold consisting of 11–stranded β–sheet surrounded by α–helices. The active site of LreRihC is formed by the residues D16, D20, D21, N45, T128, N162, E168, N170, H237 and D238. Each active site contains one Ca^2+^ ion, which is coordinated by carboxylates of three conserved aspartates (D16, D21, and D238), T128 backbone oxygen and three water molecules (Figure 6A). The subunit superposition of the LreRihC and apo forms of known RihCs (Table 1) demonstrated similar calcium coordination (Figure 6). The lack of solvent molecules coordinating calcium ions in some structures (GvaRihC, ZmaRihC, and LmaRihC) (Figure 6B–D) seems to be a result of relatively low resolution. It is also worth noting that in ZmaRihC, instead of threonine (128) residue, there is a leucine (123) residue coordinating calcium via its main chain oxygen, which, however, does not alter the coordination (Figure 6C).

To achieve insights into possible mechanisms of substrate binding by LreRihC, we further compared its active site with those from the holo form of bacterial BanRihC in complex with alpha-D-ribofuranose (Table 1, Figure 7A,B). The analysis showed the similar environment of the cavity for ribose moiety binding formed by D21 (D14 in BanRihC), T128 (T125), N162 (N160), E168 (E171), N170 (N173) and D238 (D247). It is noteworthy that the side chains of D20 and N45 in LreRihC are oriented away from the substrate-binding cavity (Figure 7A), in contrast to holo BanRihC, where the side chains of corresponding D13 and D38 might participate in substrate coordination (Figure 7B, orange dotted lines). Further structural comparison with protozoan CfaRihC in holo form (Table 1) showed that, in this case, corresponding D14 and N39 (D20 and N45 in LreRihC) also participate in ribose moiety binding (Figure 7C). However, in the CfaRihC apo form (Figure 7D), D14 is salt bridged with H241, and N39 is oriented away from the active site, which is similar to the apo form of LreRihC.

In CfaRihC residues H82, Y229 and H241 are the only residues that coordinate the nitrogenous base of the substrate [5,10]. In the case of LreRihC, H237 (H241 in CfaRihC) has a similar orientation (Figure 7A,C); however, H86 and Y226 (H82 and Y229 in CfaRihC) are located on flexible regions 3 and 2, respectively (Figure 5E–G), and thus are far away from the active site, similar to the apo CfaRihC structure. Based on structural comparison, we could speculate that these loops as well as residues D20 and N45 in LreRihC, might change their conformation upon substrate binding, as seen in known holo structures.

We further compared the structures of CfaRihC in apo (1MAS) and holo (2MAS) forms to see if the conformation of the enzyme changes significantly upon inhibitor binding, which in the case of 2MAS closely resembles the transition state (Figure 8A).

From Figure 8A, it can be seen that there are some conformational changes in the RihC structure when the substrate is bound (since the pAPIR ligand represents the transition state), but overall, the structures align well. The regions highlighted in ovals were the regions that were unresolved in our LreRihC crystal structure. Because of this, we resolved these regions and obtained a model (Figure 8B). Figure 8B shows that the unresolved regions apparently do not contain any secondary structure elements. This appears to be quite significant because when we aligned our LreRihC model and crystal structures of CfaRihC (1MAS and 2MAS) and LmaRihC (1EZR), we discovered that near to the active site, all three crystal structures have a lengthier α-spiral (Figure 8C) than LreRihC. Both CfaRihC and LmaRihC were chosen as those with kinetic parameters studied. There is also a bit of difference in Region 3 (lower oval in Figure 8C), but out of all the structures, only in 2MAS is it significantly closer to the active site. While this may be interesting in regards of conformation changes in RihC for catalysis, it may not explain the difference in catalytic properties (back in Table 2), while the differences in Region 2 (upper oval in Figure 8C) may do so.

Since only protozoan CfaRihC and LmaRihC both had their structural and kinetic parameters defined and LreRihC is a bacterial enzyme, we decided to make an amino acid alignment of all RihCs that had their kinetic parameters studied. This includes the aforementioned protozoan CfaRihC and LmaRihC as well as bacterial EcoRihC, SenRihC (both of which do not have crystal structures) and our enzyme LreRihC. The alignment is shown in Appendix A. The region that we think may explain the differences in terms of the kinetic parameters of these enzymes is located near 220–235 residues in the alignment (numbered for SenRihC), where 226 is the position wherein the LreRihC α-spiral becomes a loop region but in CfaRihC and LmaRihC the α-spiral continues making this element of the structure more rigid. It can be noticed in the alignment too since CfaRihC and LmaRihC both have more amino acid residues in that region that constitute the rest of said α-spiral. It is worth noting that both EcoRihC and SenRihC are similar to LreRihC and thus should also have a short α-spiral with a loop following after in that region. This is an interesting fact because in terms of *k_cat_* values, LreRihC is quite similar to protozoan CfaRihC, but in terms of K_M_ values, it is more like bacterial EcoRihC (the exceptions being inosine and xanthosine, which are preferred by EcoRihC but much less preferred by LreRihC). Thus, we decided to model both EcoRihC and SenRihC using AlphaFold2 and compare them to our model structure (Figure 8D). The two regions of interest are marked with yellow ovals. While it is true that residues 220–240 (upper oval) make a short α-spiral and the loop region afterwards, just like in our structure, it also can be seen that in both EcoRihC and SenRihC, this region is closer to the active site (and to be more precise, to the nucleobase-binding part of it) than in LreRihC. It is also clear that the 75–85 residues region (lower oval) of EcoRihC and SenRihC while being almost identical in secondary structure composition to that of LreRihC, is also closer to the active site cavity. The possible explanation may be that, in our enzyme, the structure may be more flexible than in EcoRihC and SenRihC (Region 3 being a bit further from the active site in LreRihC), thus the active site cavity may more easily adapt during the different parts of catalysis for the transition state forming and nucleobase leaving, and this may explain higher *k_cat_* values even for less preferred LreRihC substrates compared to other RihCs. As for inosine and xanthosine being the least preferred for LreRihC compared to other RihCs while we cannot be absolutely certain, we assume that the keto-group in position 6 of the nucleoside (hence inosine, guanosine, and xanthosine are also called 6-oxopurines) may be stabilized more poorly than the amino group in the same position for adenosine, making inosine and xanthosine much less preferrable for LreRihC.

## 3. Materials and Methods

### 3.1. Obtaining Model Protein Structures

The LreRihC structures were modeled using the free version of the AlphaFold2 (ColabFold v1.5.3) resource based on Google Collab [8,9]. Both homodimer and homotetramer were chosen as oligomerization states. For MSA (multiple sequence alignment), the mmseqs2_uniref_env mode was selected, and alphafold2_multimer_v3 was selected as the model_type. The number of recycles was 3, pairing_strategy was chosen “greedy”. The structures were visualized using the PyMOL 2.5.4 program [16].

The additional model of LreRihC containing all the missing crystal structure residues was made using the following procedure. The missing loop (DGNDQ, residues 228–232) was created manually with Discovery Studio 3.0 software suite (Accelrys, San Diego, CA, USA) using an insertion tool of residues according to the protein sequence received by MALDI-TOF. The inserted loop was refined with the “Loop refinement” protocol as implemented in Discovery Studio.

### 3.2. Obtaining the LreRihC Genetic Constructs

The RihC enzyme was cloned from the *Limosilactobacillus reuteri* LR1 bacteria (the strain was kindly provided by the All-Russian Dairy Research Institute (VNIMI)) with the addition of a fragment encoding six histidine residues sequence (His–tag) to the N- or C-terminus of the enzyme for acceleration and simplifying the enzyme purification process (hereinafter referred to as RihC_HisN for an enzyme with a His-tag at the N-terminus and RihC_HisC for an enzyme with a His-tag at the C–terminus). To clone the enzyme gene, genomic DNA of lactobacilli was isolated. For this purpose, lactobacilli cells were cultured without shaking in 20 mL of MRS medium (proteose peptone 10 g L^−1^, meat extract 10 g L^−1^, yeast extract 5 g L^−1^, glucose 20 g L^−1^, Tween–80 1 g L^−1^, ammonium citrate 2 g L^−1^, sodium acetate 5 g L^−1^, magnesium sulfate 0.1 g L^−1^, manganese sulfate 0.05 g L^−1^, and sodium hydrogen phosphate 2 g L^−1^) at 37 °C until the medium became cloudy. The cells were pelleted at 4 °C at 10,000 rpm for 5 min, suspended with 20 μL of a solution of lysozyme and RNase A in water (lysozyme concentration 50 mg mL^−1^, RNase A concentration 2 mg mL^−1^) for 20 min at 37 °C. DNA was extracted using the DNeasy mericon Food Kit (Qiagen, Germantown, MD, USA) according to the manufacturer’s protocol. The enzyme gene was cloned from *L. reuteri* genomic DNA via PCR using the two primer pairs shown below and inserted into the vector plasmid pET24a(+) containing the kanamycin resistance gene.

**LreRihC_HisN_for:** 5′—GAA GGA GAT ATA CAT ATG CAC CAC CAC CAC CAC CAC ACT ACA AAG ATT ATT ATG GAT ACT GAC CCA—3′

**LreRihC_HisN_rev:** 5′—GTG GTG GTG GTG CTC GAG TCA TTT CAT TTT GCT TAC TTC TTC TAA GAA CCA CTT—3′

**LreRihC_HisC_for:** 5′—GAA GGA GAT ATA CAT ATG ACT ACA AAG ATT ATT ATG GAT ACT GAC CCA—3′

**LreRihC_HisC_rev:** 5′—GTG GTG GTG CTC GAG TCA GTG GTG GTG GTG GTG GTG TTT CAT TTT GCT TAC TTC TTC TAA GAA CCA C—3′

The reaction mixture for PCR contained 5 μL of 10–fold buffer for Pfu DNA polymerase (200 mM Tris–HCl (pH 8.8 at 25 C), 100 mM (NH_4_)_2_SO_4_, 100 mM KCl, 1 mg mL^−1^ BSA, 1% (*v*/*v*) Triton X–100, 20 mM MgSO_4_); 5 µL of dNTP mixture (dATP, dGTP, dTTP, dCTP, each concentration 2.5 mM); 1 µL of DNA template (≈50 ng µL^−1^); 2 µL of primers (10 mM); 0.5 µL Pfu DNA polymerase (2.5 U µL^−1^) and deionized water to a total mixture volume of 50 µL. PCR was carried out in a thin-walled plastic tube with a volume of 0.2 mL (Eppendorf, Hamburg, Germany) on a BioRad T100 Thermal Cycler device (BioRad, Hercules, CA, USA). The test tube was heated for 5 min at 95 °C, and then the reaction was carried out according to the following program: denaturation—95 °C, 30 s; primer binding—54–58 °C, 1 min, chain extension—72 °C, 2 min, 25–35 cycles in total. After the last cycle, the reaction mixture was kept for an additional 10 min at 72 °C. The temperature at the second stage was chosen to be 3–5 degrees below the melting temperature of the duplexes (Tm) formed by the primers.

PCR products were purified via electrophoresis in a 1% agarose gel, followed by the isolation of DNA fragments from the gel. The purified products, as well as the selected vector pET24a(+), were treated with restriction enzymes NdeI and XhoI. The vector was selected in such a way that the enzyme gene in this vector was under the strong promoter of phage T7 RNA polymerase. To create a restriction mixture, 2 μL of 10x buffer for restriction FD Green Buffer (Thermo Scientific, Waltham, MA, USA) was mixed with 10 μL of PCR product or 2 μL of vector plasmid, then water (MiliQ grade pure) was added to bring the volume to 19 μL, and then 0.5 μL of restriction enzyme NdeI and 0.5 μL of restriction enzyme XhoI (FastDigest series, Thermo Scientific, Waltham, MA, USA) were added. The mixtures were incubated for 1 h at 37 °C and then purified in the same way as PCR products. Purified fragments after restriction were ligated using T4 DNA ligase from Thermo Scientific (Waltham, MA, USA) according to the manufacturer’s protocol. The mixtures obtained after ligation were used to transform *E. coli* DH5α cells. To control the production of the required genetic constructs, plasmid DNA was sequenced at the Center for Collective Use “Genome” (V.A. Engelhardt Institute of Molecular Biology, Russian Academy of Sciences).

Sequencing results showed that the genetic constructs contain only the RihC gene with the target insert (His-tag at the N- or C-terminus of the enzyme). The size of each of the RihC genes with a His-tag at one end or another was 927 bp, each of the genes encodes a protein with a length of 308 amino acid residues and a size of 33.38 kDa, the ascension number of the RihC enzyme from *L. reuteri* without His-tag is MBU5982057.1

### 3.3. Enzyme Expression in E. coli Cells

The resulting plasmids containing the gene encoding RihC_HisN or RihC_HisC were used to transform *E. coli* BL21(DE3) cells resistant to chloramphenicol. The pET24a(+) vector into which the RihC gene was cloned contains a kanamycin resistance gene. Transformed cells were plated on 2YT agar nutrient medium containing the antibiotics kanamycin (final concentration: 30 μg mL^−1^) and chloramphenicol (final concentration 25 μg mL^−1^) and incubated for 12–14 h at 37 °C. Next, colonies of grown bacteria were transferred to 4 mL of liquid nutrient medium 2YT containing the antibiotics kanamycin and chloramphenicol in the concentrations mentioned above and incubated for 12–14 h on a shaker at 30 °C and 180 rpm. After this, 20 μL of cells was transferred into culture flasks in 20 mL of 2YT liquid nutrient medium containing the antibiotic kanamycin at the concentration mentioned above and incubated at 37 °C and 180 rpm until absorbance at 600 nm A_600_ = 0.8–1. Next, the entire volume of the flasks was transferred to large flasks for cultivation in 180 mL of 2YT liquid nutrient medium without antibiotics, and the cells were grown at 30 °C and 100 rpm until absorption A_600_ = 0.8–1 was achieved. Once the desired absorption was achieved, cells were induced with iPTG (final concentration in the flask 0.1 mM), and then flasks were incubated at 20 °C and 120 rpm for 16–18 h. Next, the flasks were removed from the shaker, the cell suspension was sedimented in an Eppendorf 5804 R centrifuge at 4 °C and 5000 rpm, and the supernatant was drained. Cells were resuspended in a buffer containing 0.05 M Tris–HCl, 0.5 M NaCl, 0.02 M imidazole pH 7.5 (buffer A) in a ratio of 4:1 by weight (four parts of buffer per one part of cells). The resulting cell suspensions were frozen at −20 °C until purification.

### 3.4. Enzyme Purification

*E. coli* cells containing the RihC enzyme were destroyed on an ultrasonic disintegrator. The cell debris was settled on an Eppendorf 5804 R centrifuge at 4 °C and 6000 rpm, and the supernatant was transferred into clean plastic tubes with a volume of 50 mL. Purification was carried out on an AKTA Start chromatography system using a HisTrap FF 1 mL column (Cytiva, Marlborough, MA, USA). The column was pre-equilibrated with buffer A. The supernatant after cell disruption was applied to the column at a flow of 0.5 mL min^−1^. After the disappearance of the absorption peak at 280 nm of impurity proteins that did not bind to the column, the target enzyme was eluted with a linear gradient of increasing concentration of buffer B (0.05 M Tris–HCl, 0.5 M NaCl, 0.5 M imidazole pH 7.5). The resulting enzyme solution was desalted using size-exclusion chromatography on a Sephadex G25 column into a solution containing 0.05 M Tris–HCl pH 7.5. The purity of the resulting enzyme solution was confirmed via SDS–PAGE as described in [17].

Confirmation of the amino acid sequence of the RihC was performed using MALDI mass spectrometry. When conducting these studies, the equipment of the Center for Collective Use “Industrial Biotechnologies” of the Federal State Institution “Federal Research Center “Fundamental Foundations of Biotechnology” of the Russian Academy of Sciences” was used. After Coomassie Brilliant Blue staining of the SDS-PAGE gel was carried out, a piece of the gel containing the enzyme (3–4 mm^3^) was cut out, washed twice with 100 μL of 40% acetonitrile in 0.1 M NH_4_HCO_3_ for 20 min at 37 °C to remove the dye, and transferred to 100 μL of acetonitrile for dehydration. After dehydration, the gel piece was dried to remove acetonitrile and 3.5 μL of modified trypsin solution (Promega, Madison, WI, USA) in 0.05 M NH_4_HCO_3_ was added. Hydrolysis was carried out for 20 h at 37 °C; next, 5.25 μL of 0.5% trifluoroacetic acid (TFA) in 50% aqueous acetonitrile solution was added to the reaction mixture and thoroughly mixed. The resulting supernatant was used for MALDI mass spectroscopy analysis. The samples for mass spectrometry were prepared by mixing 1.5 μL of the tryptic hydrolysate with 0.5 μL of 2,5 dihydroxybenzoic acid (10 mg mL^−1^ in 20% aqueous acetonitrile, 0.5% TFA; Sigma–Aldrich, Burlington, MA, USA). The resulting mixture was air-dried. Mass spectra were recorded with an Ultraflextreme MALDI-TOF/TOF mass spectrometer (Bruker, Bremen, Germany) equipped with a positive-ion UV laser and a reflectron. The accuracy of the measurement of monoisotopic masses after recalibration with the peaks for trypsin autolysate was 0.002–0.011% (20–110 ppm). The spectra were obtained within the mass range of 500–6500 m z^−1^; laser power was selected to achieve the best resolution. Proteins were identified using Mascot software (Matrix Science Ltd, Boston, MA, USA; www.matrixscience.com (accessed on 10 August 2023)). Mass spectra were processed with the FlexAnalysis 3.3 software package (Bruker, Bremen, Germany). Using the Mascot program (“peptide fingerprint” option), we searched a local database with the above accuracy, taking into account the following possible modifications: acetylation (protein N-terminus), Gln pyroGlu (N-terminal Q), oxidation (M), propionamide (C). Candidate proteins with the confidence score > 42 in the NCBI database were considered reliably identified (*p* < 0.05).

### 3.5. Determination of the Oligomeric Composition of the Enzyme

The oligomeric composition of the enzyme was determined using analytical size-exclusion chromatography (gel filtration) on an AKTA Start chromatograph on a HiLoad 16/600 Superdex 200 pg column (Cytiva, Marlborough, MA, USA). To construct a calibration curve, a set of molecular weight standards for gel filtration and analytical electrophoresis Gel Filtration Calibration Kit HMW (Cytiva, Marlborough, MA, USA) was used. Column equilibration and calibration were performed according to the manufacturer’s protocol. The test sample was added to the column in an amount of 1 mg, and its retention time on the column was observed. Next, using a previously constructed calibration curve, the molecular mass of the sample was determined and compared with the theoretical mass of the monomer, from which a conclusion was drawn about the oligomeric composition of the enzyme.

### 3.6. Carrying Out an Enzymatic Reaction

The enzymatic reaction was carried out using 99% pure uridine, citidine, adenosine, guanosine, inosine, xanthosine, 2′–deoxyuridine, thymidine, vidarabine or 5–methyluridine from Sigma–Aldrich (Burlington, MA, USA) and water purified using a Milli-Q unit (Merck, Darmstadt, Germany). The reaction was carried out as follows: the desired stock concentration of nucleoside was prepared by weighing it and dissolving it in Tris–HCl buffer pH 7.5, then 500 μL of said nucleoside at the required concentration (diluted with water when needed) was added to plastic test tubes (total volume 1.5 mL), and five replicates were made for each final concentration of nucleoside. The number of points on the curve (i.e., the number of nucleoside samples) depends on the further accuracy of plotting the reaction rate versus substrate concentration curve; in this work, it was 15 points for each substrate where applicable. Next, 5 μL of purified RihC solution at a concentration of approximately 200 μg mL^−1^ was added to each sample (the final concentration in the solution is 2 μg mL^−1^) and stirred. At certain time intervals, the enzymatic reaction was stopped by adding 5 µL of concentrated HCl (time differed for different nucleosides). The prepared samples were then used for analysis.

### 3.7. Conducting the Analysis Using HILIC

For analysis, 99% pure uracil, cytosine, adenine, guanine, thymine, xanthine, or hypoxanthine from Sigma–Aldrich (Burlington, MA, USA), water purified using a Milli–Q unit (Merck, Darmstadt, Germany), and HPLC gradient grade acetonitrile (AppliChem, Darmstadt, Germany) were used. A detailed description of the development of this approach to enzymatic activity measurements is discussed in [12]. The experiments were carried out using the HPLC system consisting of a Dionex3000 Ultimate chromatograph (Thermo Fisher Scientific, Waltham, MA, USA) with a two-channel gradient pump, an automatic sample injection system, a column thermostat, and a diode array spectrophotometric detector. Chromatograms were recorded using a personal computer and the Chromeleon 7 software package (Thermo Fisher Scientific, Waltham, MA, USA). The separation of nucleobase and nucleoside was carried out using a mixture of 20 mM ammonium acetate buffer, pH 4.7 and acetonitrile (10:90 vol.%) as the mobile phase, the eluent flow rate was 1 mL min^−1^, the thermostat temperature was 30 °C, and UV detection was performed at 254 nm. Before starting the analysis, a series of solutions of nucleobase in water with known concentrations was prepared. All samples (both nucleobase and corresponding nucleoside samples from the previous paragraph) were added to clean vials, 100 μL each, then 900 μL of acetonitrile was added to each. The calibration plot was created using five concentration levels of nucleobase. Each point of the calibration plot was the average of three peak area measurements. The proposed method was also validated for limits of detection, linearity and intra-day and inter-day precision, as shown in [12]. A set of “blank” injections of a nucleoside solution of a certain concentration was also made to check for corresponding nucleobase impurities. After that, the analyzed samples were injected into the chromatograph, and the peak areas of nucleobase and nucleoside were recorded; each sample was injected three times to check the data reproducibility. The peak areas of nucleobase were subsequently converted into concentrations using a calibration plot. The obtained data were used to create the dependence of the enzymatic reaction rate on the substrate concentration, from which the kinetic parameters of this reaction were determined using the Origin Pro 8.5 program (OriginLab, Northampton, MA, USA).

### 3.8. Kinetic Parameters Determination for RihC Enzyme

The dependence of the rate of the enzymatic reaction on the substrate concentration was obtained as described above. This dependence was analyzed using Origin Pro 8.5. The Michaelis constant K_M_ was determined using nonlinear regression in the range of substrate concentrations of 0.5–5K_M_. From the same dependence, the maximum speed of the enzymatic reaction V_max_ was determined. The concentration of the purified enzyme was determined via the absorption of the protein solution at 280 nm using the calculated extinction coefficient according to the formula: C_protein_ (mg mL^−1^) = A_280_/εl, where l is the optical path length equal to 1 cm. The absorption of the enzyme solution was measured in quartz cuvettes with an optical path of 10 mm on a Schimadzu UV1800 PC spectrophotometer. Using V_max_ and C_protein_, the rate constant of the enzymatic reaction *k_cat_* was calculated from the equation V_max_ = *k_cat_**C_protein_.

### 3.9. Study of Enzyme Thermostability

To study the thermal stability of the RihC enzyme, differential scanning calorimetry (DSC) was used. The temperature stability study was carried out using a Nano DSC differential adiabatic scanning microcalorimeter (TAInstruments, New Castle, DE, USA). The working volume of capillary calorimetric platinum cells was 300 μL. To prevent the formation of bubbles and boiling of solutions when the temperature increased, an excess pressure of 3 atm was maintained in the calorimeter cells. Before the experiment, instrumental baseline values were determined and then subtracted from the data obtained for the protein. During measurements, a buffer solution was placed in the control cell, and a solution of the enzyme under study in the same buffer solution was placed in the working cell. 0.1 M NaPB pH 7.0 was used as a buffer solution. The enzyme concentration was 1–2 mg mL^−1^, and the heating rate was 1 °C min^−1^.

Additionally, the thermal inactivation kinetics for RihC were also studied. The temperature stability of the enzyme was measured in 0.1 M sodium phosphate buffer, pH 7.0, at several temperatures in the range of 45–60 °C. For each experiment, a series of 0.5 mL plastic tubes were prepared with 100 µL of enzyme solution (0.2 mg mL^−1^) in each. The test tubes were placed in a water thermostat preheated to the required temperature (temperature control accuracy ±0.1 °C). At certain points in time, one tube was taken and transferred on ice for 5 min, after which the tube was centrifuged for 3 min at 12,000 rpm in an Eppendorf 5415D centrifuge. Residual enzymatic activity was measured as described above. The thermal inactivation rate constant *k_in_* was determined as the slope of the straight line from the natural logarithm of the residual activity versus time plot (semilogarithmic coordinates ln(A/A_0_) − t) via linear regression using the OriginPro 8.5 program (OriginLab, Northampton, MA, USA).

### 3.10. Crystallization and Data Collection

Initial crystallization screening was performed on a robotic system (Rigaku Americas Corporation, The Woodlands, TX, USA) using 96-well VDX plates (Hampton Research, Aliso Viejo, CA, USA) and commercial crystallization screens from Hampton Research (Aliso Viejo, CA, USA) and Molecular Dimensions Inc (Holland, OH, USA) via the “sitting drop” vapor diffusion method. A 10 mg/mL sample of the RihC in 20 mM Tris–HCl buffer pH 8.0 was mixed with the crystallization solution in the ratios of 1:1, 1:2 and 2:1 (0.3 μL drop volume). The volume of the precipitant solution in the reservoir was 50 µL. The crystallization was observed under the following conditions: 0.2 M Magnesium acetate tetrahydrate, 0.1 M Sodium cacodylate trihydrate pH 6.5, 20% *w*/*v* Polyethylene glycol 8000 at 1:1 ratio at 288 K.

The RihC crystal was briefly soaked in a mother liquor containing 20% glycerol immediately before diffraction data collection and flash-frozen in liquid nitrogen. Datasets were collected at 100 K at Rigaku OD XtaLAB Synergy–S (Rigaku, Tokyo, Japan). The datasets were indexed, integrated, and scaled using the XDS package [18]. Space groups were suggested by Pointless [19] as P12_1_1 (Table 4).

### 3.11. Structure Solution and Refinement

The structure of the RihC was solved by using the molecular replacement method using the MOLREP program [20] with the atomic coordinates of the *E. coli* RihA pyrimidine nucleosidase (PDB ID: 3G5I) as a starting model. Four copies of the protein were found in an asymmetric unit. The refinement of the structure was carried out using the REFMAC5 program of the CCP4 suite (Harwell Science and Innovation Campus, Didcot, UK) [21]. The isotropic B-factor and the hydrogen atoms in fixed positions were included during the refinement. The manual rebuilding of the model was carried out using the COOT interactive graphics program [22]. It is noteworthy that in the final model, residues 228–232 have poor electron density and were thus not modeled. Additionally, residues 86 (chain A) and 84–86 (chain D) were not modeled for the same reason.

### 3.12. Structure Analysis and Validation

The visual inspection of the modeled structure was carried out using the COOT program and the PyMOL Molecular Graphics System, Version 4.6 (Schrödinger, New York, NY, USA). The structure comparison and superposition were made using the PDBeFOLD (EMBL-EBI, Hinxton, UK) program [23]. The contacts were analyzed using PDBePISA (EMBL-EBI, Hinxton, UK) [24].

## 4. Conclusions

Thus, we successfully obtained RihC from *Limosilactobacillus reuteri* LR1 bacteria and demonstrated that the N-terminal His-tag form expressed approximately three times better than the corresponding C-terminal form.

The activity of the enzyme was measured towards a wide range of substrates, including ribonucleosides: adenosine, guanosine, cytidine, uridine, inosine, xanthosine, 5-methyluridine, as well as with deoxyribonucleosides: thymidine and 2′-deoxyuridine and one arabinoside vidarabine. It was shown that the reaction occurs only with ribonucleosides and arabinoside. For uridine, cytidine, adenosine, inosine, and xanthosine, the kinetic parameters K_M_ and *k_cat_* for each form of the enzyme were determined. It was shown that the Michaelis constants for the enzyme are in good agreement with those for other RihC enzymes, and the turnover rate constants for this enzyme were quite high, even for less preferred substrates.

The thermal stability of LreRihC_HisN was studied using the kinetics of thermal inactivation of the enzyme, as well as using DSC. Finally, the crystal structure of the enzyme was obtained at 1.9 Å resolution, and its comparison with those from other organisms shed light on the structural features of the enzyme’s active site.

## Figures and Tables

**Figure 1 ijms-25-00538-f001:**
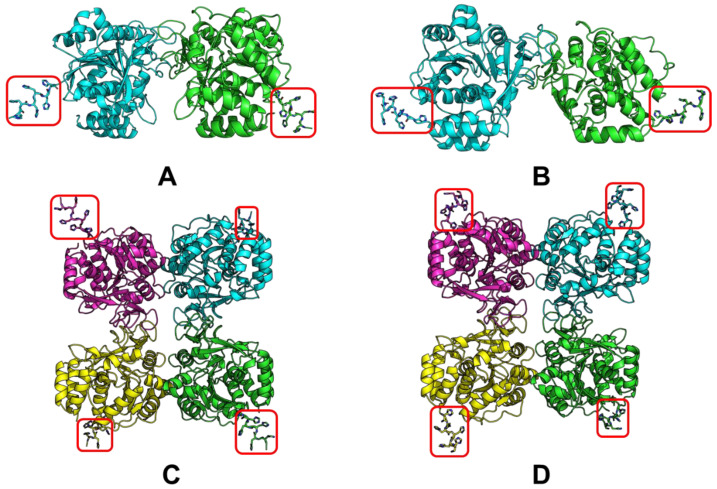
LreRihC models made with AlphaFold2. (**A**) LreRihC_HisN dimer, (**B**) LreRihC_HisC dimer, (**C**) LreRihC_HisN tetramer, (**D**) LreRihC_HisC tetramer. His-tag is shown in red.

**Figure 2 ijms-25-00538-f002:**
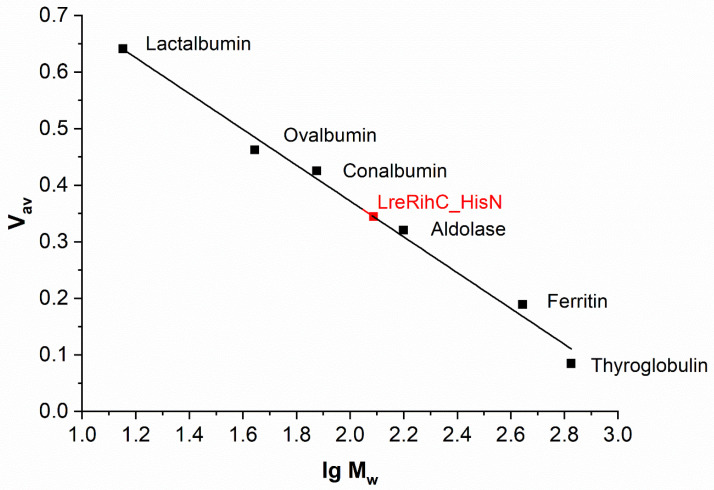
The calibration curve of analytical gel filtration. LreRihC_HisN is shown in red.

**Figure 3 ijms-25-00538-f003:**
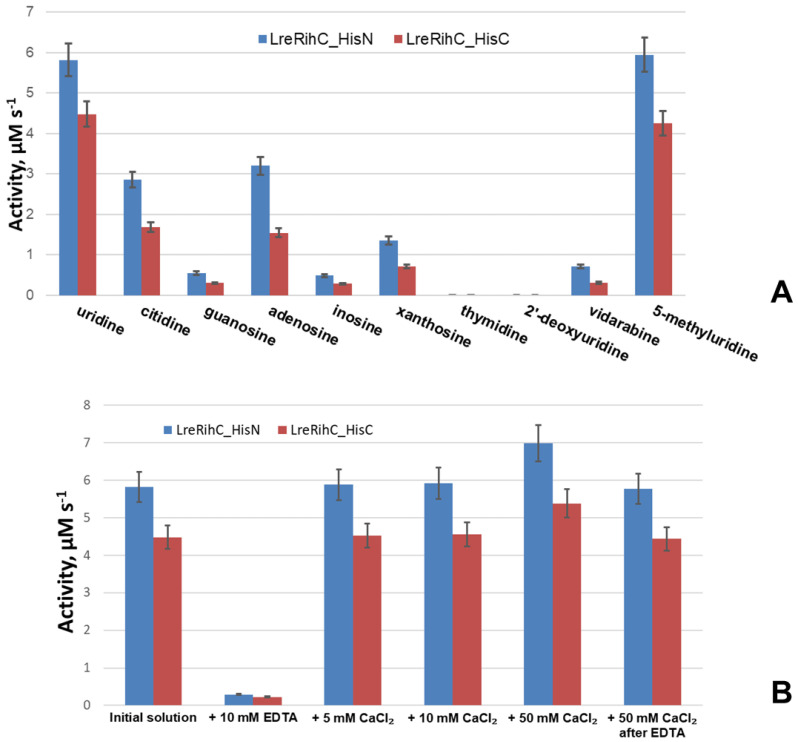
(**A**) LreRihC activity with different substrates at 2 mM for each substrate. (**B**) LreRihC activity with 20 mM of uridine after the addition of calcium ions or EDTA to the enzyme in different concentrations. LreRihC_HisN activity is shown in blue, LreRihC_HisC activity is shown in red.

**Figure 4 ijms-25-00538-f004:**
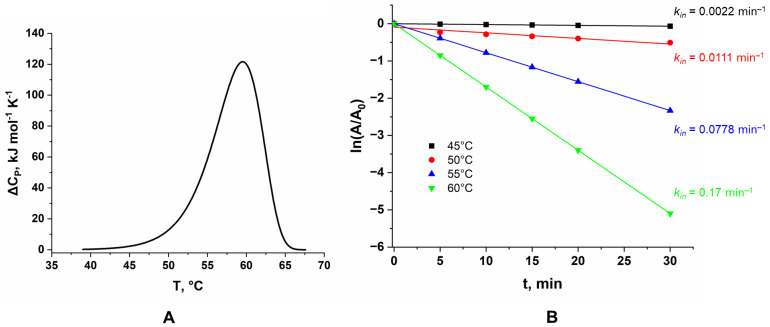
(**A**) DSC curve for LreRihC_HisN, (**B**) dependence of the residual activity of the LreRihC_HisN on incubation time for different temperatures in the reaction with 30 mM of uridine.

**Figure 5 ijms-25-00538-f005:**
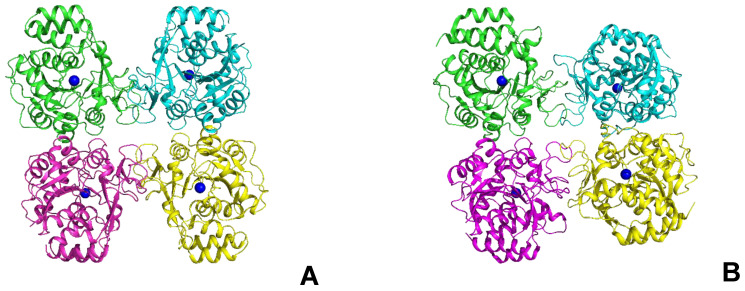
Comparison of RihCs crystal structures. Functional tetramers of LreRihC (**A**) and LmaRihC (**B**) as well as functional dimers of GvaRihC (**C**) and ZmaRihC (**D**) (Table 1). Proteins are colored by chains. Green subunits are presented in the same orientation. Ca^2+^ ions are shown as blue spheres to assess subunit orientation. Superposition of the subunits of LreRihC (gray) with LmaRihC (orange, (**E**)), GvaRihC (magenta, (**F**)) and ZmaRihC (red, (**G**)). Well-fitted regions are shown semi-transparent for clarity.

**Figure 6 ijms-25-00538-f006:**
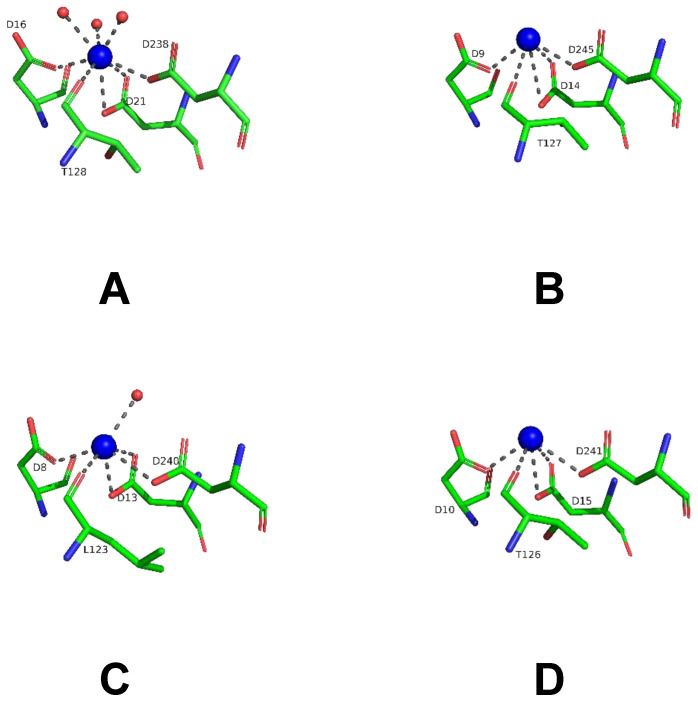
Coordination (gray dotted lines) of Ca^2+^ ion (blue spheres) in the active sites of the LreRihC (**A**), GvaRihC (**B**), ZmaRihC (**C**) and LmaRihC (**D**).

**Figure 7 ijms-25-00538-f007:**
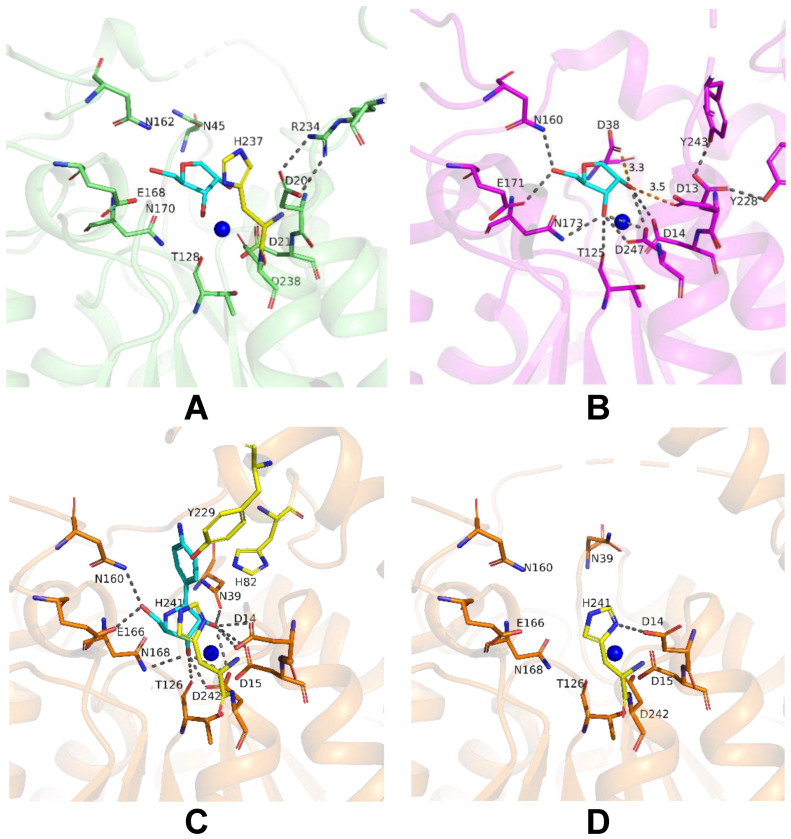
Active site of LreRihC shown in green (**A**) and BanRihC shown in magenta (**B**). A ligand from superimposed BanRihC structure is shown in cyan on both panels. Active sites of CfaRihC in complex with inhibitor pAPIR (shown in cyan) (**C**) and in its apo form (**D**) are shown in orange and in the same orientation. Residues that coordinate the nitrogenous base of the substrate are colored yellow. Hydrogen bonds and salt bridges are shown as gray dotted lines, and important distances are labeled and shown in orange. Calcium ion on panels (**A**–**C**), and potassium ion on panel (**D**) are shown as blue spheres.

**Figure 8 ijms-25-00538-f008:**
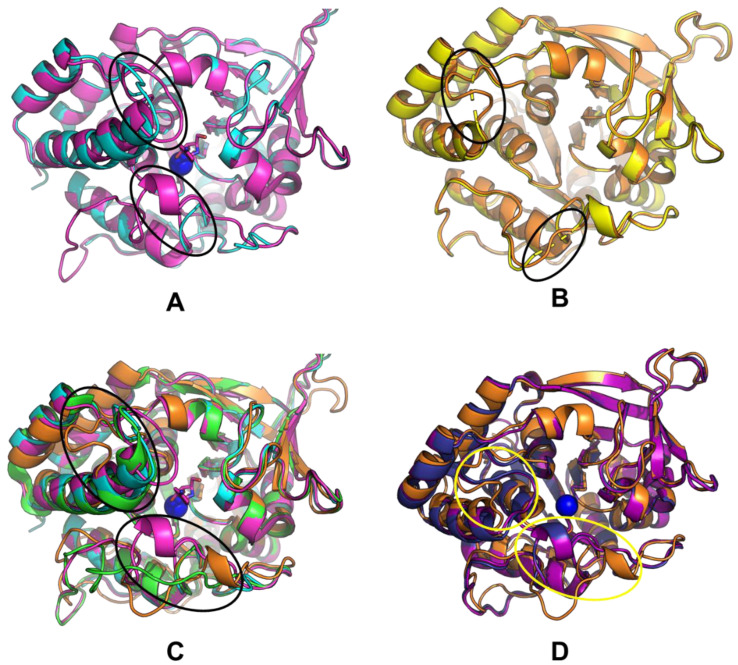
(**A**) Comparison of CfaRihC crystal structures 1MAS and 2MAS, (**B**) comparison of LreRihC crystal structure and computed model with resolved regions, (**C**) comparison of LreRihC model, CfaRihC (1MAS and 2MAS), and LmaRihC (1EZR), (**D**) comparison of LreRihC model, EcoRihC computed model, and SenRihC computed model. 1MAS is shown in cyan, 2MAS is shown in magenta, 1EZR is shown in green, LreRihC crystal structure is shown in yellow, LreRihC computed model is shown in orange, EcoRihC computed model is shown in dark blue, SenRihC computed model is shown in purple, calcium ion is shown as a blue sphere, pAPIR from 2MAS is shown in magenta. For (**A**,**C**,**D**) amino acid regions 220–240 (Region 2 as discussed for Figure 5) and 75–85 (part of Region 3 as, discussed for Figure 5) are highlighted with ovals (upper oval for Region 2 and lower oval for Region 3).

**Table 1 ijms-25-00538-t001:** Structural comparison of RihC hydrolases deposited in the PDB database. Abbreviations used below are shown in brackets. Superposition was made with RihC from Limosilactobacillus reuteri. * Current paper.

Organism Type	Organism	PDB Code	Resolution	RMSD, Å	Sequence Identity, %	Percentage of the Aligned Residues, %
Bacteria *	*Limosilactobacillus reuteri* LR1(LreRihC) *	8QND	1.9	-	-	-
Bacteria	*Bacillus anthracis* (BanRihC)	2C40 (holo form)	2.2 Å	1.23	24	87
Bacteria	*Gardnerella vaginalis* 315–A (GvaRihC)	6BA1 (apo form)	2.9 Å	1.43	32	87
Plants	*Physcomitrella patens* (PpaRihC)	4KPN (apo form)	3.35 Å	1.17	32	83
Plants	*Zea mays* (ZmaRihC)	4KPO (apo form)	2.49 Å	1.12	33	87
Protozoa	*Crithidia fasciculate* (CfaRihC)	1MAS (apo form)	2.5 Å	1.02	34	91
		2MAS (holo form)	2.3 Å	1.06	34	91
Protozoa	*Leishmania braziliensis* (LbrRihC)	5TSQ (holo form)	1.53 Å	1.02	33	87
Protozoa	*Leishmania major* (LmaRihC)	1EZR (apo porm)	2.5 Å	1.04	34	91

**Table 3 ijms-25-00538-t003:** List of abbreviated RihC names.

Organism Type	Organism Name	Shortened RihC Name
Bacteria	*Bacillus anthracis*	BanRihC
Bacteria	*Escherichia coli*	EcoRihC
Bacteria	*Gardnerella vaginalis* 315-A	GvaRihC
Bacteria	*Limosilactobacillus reuteri* LR1	LreRihC
Bacteria	*Salmonella enterica*	SenRihC
Plants	*Physcomitrella patens*	PpaRihC
Plants	*Zea mays*	ZmaRihC
Protozoa	*Crithidia fasciculata*	CfaRihC
Protozoa	*Leishmania braziliensis*	LbrRihC
Protozoa	*Leishmania major*	LmaRihC

**Table 4 ijms-25-00538-t004:** Data collection and refinement statistics.

Diffraction Source	Institute of Organic Chemistry RAS (Rigaku OD XtaLAB Synergy–S)
Wavelength (Å)	1.54
Temperature (K)	100
Detector	HyPix–6000HE
Crystal–to–detector distance (mm)	32.0
Rotation range per image (°)	0.35
Total rotation range (°)	344.75
Space group	P12_1_1
a, b, c (Å)	84.11, 81.53, 86.85
α, β, γ (°)	90.0, 95.8, 90.0
Average mosaicity (°)	0.27
Resolution range (Å)	63.38–1.90 (1.93–1.90)
Completeness (%)	99.8 (100)
Average redundancy	6.4 (6.6)
I/σ(I)	19.5 (5.5)
Rmeas (%)	9.1 (37.7)
CC_1/2_	91.4 (95.6)
R_fact_ (%)	16.9
R_free._ (%)	20.1
RMSD Bonds (Å)	0.01
RMSD Angles (°)	1.85
Ramachandran favored (%)	96.8
Ramachandran allowed (%)	3.1
PDB entry code	8QND

## Data Availability

The data presented in this study are available on request from the corresponding author.

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
