# Peer review of "Structure–Functional Examination of Novel Ribonucleoside Hydrolase C (RihC) from Limosilactobacillus reuteri LR1"

_ijms, 2023, doi:10.3390/ijms25010538_

Round 1

Reviewer 1 Report

Comments and Suggestions for Authors

Author Response

Reviewer 1 - Answers

The manuscript "Structure-functional examination of novel ribonucleoside hydrolase C (RihC) from Limosilactobacillus reuteri LR1" describes the cloning, expression, characterization, and determination of the crystal structure of LreRihC. The article is an excellent work in the field. I've noticed only several minor issues.

  1. I tried to access 8QND in the protein databank. It was not accessible, as access would be available after the publication. However, I noticed that some of the authors (depositors) of the crystal structure in the protein databank are not listed as authors in the article.

Thank you for reviewing our manuscript! The deposition process of the structure to PDB is nearly finished and the structure will be released on 20th of December. Moreover, we attach the .pdb file of our structure with the revised manuscript so the reviewers could see it. As for the other point, as you noted some of the deposition authors are not in the list of the manuscript authors as it is our standard laboratory policy.

  1. In the material and method sections, it is described that the enzymatic reaction is performed in water. Typically, the enzymatic reaction is performed in buffer solutions. I wonder if there is a reason for that, or maybe there is an omission in the materials and method section?

Thank you for your comment. This was an oversight on our part. All enzymatic reactions were performed in Tris- HCl buffer. We’ve made the necessary corrections in the text.

  1. Calcium is necessary for the catalysis of LreRihC. However, several aspects of this issue are confusing. In the materials and methods, it is mentioned that the enzymatic reactions are performed in water (e.i., in the absence of calcium). Does this mean that the Km values shown in Table 3 are determined in the absence of calcium? The Lre RihC protein is purified using a buffer that not only does not contain calcium but contains EDTA. Then how is this protein catalytically active in water? And how is calcium present in the crystal structure of the protein?

We apologize for this error. Enzyme was purified into Tris-HCl buffer without EDTA at the last step. We’ve made the necessary corrections to that part. As for the calcium ion we did not add additional calcium to the enzyme when we conducted kinetic studies of our enzyme however as we noted later in the manuscript, we’ve measured enzyme activity with the addition of calcium in different concentrations and did not get significant changes in enzyme activity. We assume that most of the calcium ions that our enzyme has come from cultivation medium during the expression of this enzyme and thus when we’ve added additional calcium it had almost no effect on activity of our enzyme.

  1. In figure 3, the highest concentrations are likely non-physiological. In addition to allosteric mechanisms, other possible explanations could be non-specific binding or aggregation (nucleosides may form hydrogen bonds with themselves or the protein at high concentrations). The Km values are determined based on the curves presented in supplementary data (lower concentrations) and could be more apparent if they are moved in a separate figure in the main text before Table 3. Otherwise, it appears that the Km values were derived from the curves in Figure 3A, B, which, of course, is not possible.

Thank you for your comment. We made necessary adjustments to our figures to make it more clear which curves we used for determining Km values. We also thank you for your comment on possible explanations for the effects we’ve got for high uridine and cytidine concentrations.

We would also like to sincerely apologize to the reviewer for the mistake in our calculations that we haven’t noticed before. When we conducted enzymatic reactions, we first diluted our enzyme 10 times and then used this solution for the reaction itself. However, when we added 5 µL of the enzyme to the 500 µL reaction solution our enzyme was also diluted 100 times additionally. When we calculated kcat values for all of the substrate reactions we forgot to take into account this 100 times dilution of our enzyme and thus our kcat values were 100 times lower than they should have been. We’ve carefully checked and recalculated all of our data for this revision and now present the correct kcat values in our revised manuscript. We’ve also rewritten the appropriate parts of kinetic and structural discussion sections.

Reviewer 2 Report

Comments and Suggestions for Authors

This is a very well designed work on crystalline study of the enzyme. However there are two points should be resolved prior the publication.

1. Figure 3B. Please explain drop in the rate of enzymatic reaction for concentrations 40, 50 and 100 mM.

2. In the manuscript there is computational structures on enzyme compared to structures from PDB and structures obtained by authors. However it is not obvious where is which one. Please make additional descriptions in figures.

Author Response

Reviewer 2 Answers

This is a very well designed work on crystalline study of the enzyme. However there are two points should be resolved prior the publication.

  1. Figure 3B. Please explain drop in the rate of enzymatic reaction for concentrations 40, 50 and 100 mM.

Thank you for reviewing our manuscript! As we’ve tried to discuss this fact in the manuscript, we assume that for high concentrations of cytidine allosteric inhibition of the enzyme occurs and the inhibitory effect is strong. However, we cannot firmly claim that this is true, this is only our assumption on unusual behavior of our enzyme for cytidine and uridine at high substrate concentrations.

Due to this part leaving most reviewers confused we decided to remove this since it has no significant value to our article and we haven’t investigated it thoroughly. This can be a material for the future research on this enzyme.

  1. In the manuscript there is computational structures on enzyme compared to structures from PDB and structures obtained by authors. However it is not obvious where is which one. Please make additional descriptions in figures.

Thank you for your comment. We made things clearer.

We would also like to sincerely apologize to the reviewer for the mistake in our calculations that we haven’t noticed before. When we conducted enzymatic reactions, we first diluted our enzyme 10 times and then used this solution for the reaction itself. However, when we added 5 µL of the enzyme to the 500 µL reaction solution our enzyme was also diluted 100 times additionally. When we calculated kcat values for all of the substrate reactions we forgot to take into account this 100 times dilution of our enzyme and thus our kcat values were 100 times lower than they should have been. We’ve carefully checked and recalculated all of our data for this revision and now present the correct kcat values in our revised manuscript. We’ve also rewritten the appropriate parts of kinetic and structural discussion sections.

Reviewer 3 Report

Comments and Suggestions for Authors

·      The manuscript is comprehensive and clear; however, I have noted some revisions. The importance of ribonucleoside hydrolase C is explained in detail and the necessity of this study is emphasized. The study was carried out both experimentally and structural analysis applications were made in a virtual environment. Virtual structure analysis studies should also be mentioned in the abstract and keywords related to this subject should be added.

·       It was mentioned in the manuscript that MALDI/TOF analysis was performed. The results of this analysis were not included in the article, and the analysis results were not presented as supplementary file.

·       In the material method section which was explained enzyme expression methods, the authors only cited their own previous articles as references and did not refer to other similar articles.

·       It is unclear why E. coli is preferred for enzyme expression.

   By using more current references and comparing it with similar articles, the originality of this study can be demonstrated and the discussion section can be strengthened.

Author Response

Reviewer 3 Answers

The manuscript is comprehensive and clear; however, I have noted some revisions. The importance of ribonucleoside hydrolase C is explained in detail and the necessity of this study is emphasized. The study was carried out both experimentally and structural analysis applications were made in a virtual environment. Virtual structure analysis studies should also be mentioned in the abstract and keywords related to this subject should be added.

Thank you for reviewing our manuscript! We made necessary changes suggested by this comment.

It was mentioned in the manuscript that MALDI/TOF analysis was performed. The results of this analysis were not included in the article, and the analysis results were not presented as supplementary file.

Thank you for your comment. We provided MALDI results in supplementary files as Figures S2 and S3.

In the material method section which was explained enzyme expression methods, the authors only cited their own previous articles as references and did not refer to other similar articles.

Thank you for your comment. However, “Enzyme expression” section does not contain any citations whatsoever because the method of expression was developed in our laboratory and is thoroughly given in this section. As for “Enzyme purification section” we used the newly developed in our laboratory approach for fast SDS-PAGE so we cited the work about it and as for MALDI we’ve now removed the citation and instead given the details on the process.

It is unclear why E. coli is preferred for enzyme expression.

  1. coli is a great model organism that can grow quite quickly and can yield large amounts of recombinant enzyme(s). This organism is also quite easy to work with be it cultivation or destruction (for protein purification). For these points we prefer using E. coli as our recombinant system.

By using more current references and comparing it with similar articles, the originality of this study can be demonstrated and the discussion section can be strengthened.

Thank you for your comment. If we talk about RihC or even other Rih hydrolases sadly there is no recent experimental studies (as in studies performed in recent five years) and we’ve referenced the newest and most notable ones in this regard.

We would also like to sincerely apologize to the reviewer for the mistake in our calculations that we haven’t noticed before. When we conducted enzymatic reactions, we first diluted our enzyme 10 times and then used this solution for the reaction itself. However, when we added 5 µL of the enzyme to the 500 µL reaction solution our enzyme was also diluted 100 times additionally. When we calculated kcat values for all of the substrate reactions we forgot to take into account this 100 times dilution of our enzyme and thus our kcat values were 100 times lower than they should have been. We’ve carefully checked and recalculated all of our data for this revision and now present the correct kcat values in our revised manuscript. We’ve also rewritten the appropriate parts of kinetic and structural discussion sections.

Reviewer 4 Report

Comments and Suggestions for Authors

In this article, Shaposhnikov et al describe the kinetic and structural characterization of a ribonucleotide hydrolase (RihC) from the lactobacilli Limosilactobacillus reuteri. Using a newly developed HPLC based kinetics assay, the authors initially describe the detailed kinetic characterization of RihC against a variety of ribonucleosides and deoxyribonucleosides, finding that RihC requires the 2’OH to catalyze their reactions. Kinetic conclusions are however muddled by complex product inhibition by one product, cytidine, and extrapolation of a single point into larger conclusions about substrate activation with uridine. Conclusions are also made about kinetic differences based on the location of a histag. However, given the distant location of either terminus from active site and the identical KM for the N-terminal and C-terminal histag variants, these shifts in kinetic activity are more likely due to minor variations in enzyme concentrations and do not reflect intrinsic effects of histag location on catalytic activity. The authors really build a more complete story with the determination of the structure of RihC and detailed comparison to similar homologues. Through these structural comparisons, the authors build a compelling story about small structural shifts in dynamic loops that provide differential catalytic activity in RihC versus homologues. With the revisions outlined below and additional documents prepared for reviewers, this article will be prepared for publication in IJMS.

Lack of necessary material for review:

1.     The authors have not provided the necessary access for the reviewers to a preprint of a submitted article on their kinetic assay, which is essential to interpreting their kinetic results. They also did not provide review access to their submitted but not released PDB file so that reviewers can confirm the proper structure and structural variables for their submission.

Revisions to article:

2.     Issues with kinetic results and interpretation

a.     Figure 3B for kinetics with cytidine indicates strong product inhibition in the measured kinetics. The authors should analyze this product inhibition and include this product inhibition in their kinetic analysis.

                                               i.     For fitting kinetic data with product inhibition, see Variot, Cillian, et al. "Mapping roles of active site residues in the acceptor site of the PA3944 Gcn5‐related N‐acetyltransferase enzyme." Protein Science 32.8 (2023): e4725.

b.     Figure 3A for kinetics with uridine. The authors make multiple conclusions about the increase in the rate at the highest concentration of uridine and state that uridine may lead to enzyme activation at high concentrations.

                                               i.     This conclusion is not supported by the data as this change is only reflected in a single point presented without error values. Additionally, no other points are presented between 50 and 100 mM showing that this trend is linear and shifts over concentration. If the authors want to make this conclusion, they need to conduct additional kinetics between 50-100 mM and present their kinetic values with errors shown to confirm that these shifts are outside the error of the measurements.

1.     Even with these additional measurements, this substrate activation is unlikely as the authors did not present evidence for a potential allosteric site within their structure of RihC that would be required for this substrate activation.

c.     Figure 4 and Table 3. The authors present a large amount of kinetic data showing that the N-terminal histag version of RihC is higher in activity than the C-terminal histag version of RihC. The authors interpret these shifts as representing an intrinsic impact of the histag on the kinetic properties of RihC. However, a simpler explanation that fits more clearly with their data is that the concentrations for these two histag versions are off slightly and this skews the relative data for only the kcat values for C-terminal histag version. The authors need to revise the overarching conclusions about the histag’s impact on measured kinetics.

                                               i.     When looking at the kinetic data in Figure 4, the relative difference between the N-terminal and C-terminal data stays constant and indicates a systematic difference between the concentration of these proteins.

                                             ii.     Additionally, in Table 3, the KM values for every substrate with the C-terminal and N-terminal histag versions are nearly identical, indicating that the only change in the kinetics is in the Vmax/kcat values, which is dependent on the enzyme concentration, and that the enzyme concentration is a more likely explanation for these observed differences in kcat values.

                                            iii.     Their own structural analysis (Figure 1) shows that the two termini are located nearby in space and far distant from the active site, indicating that histag placement is unlikely to impact the properties of RihC. Histag placement also has no impact on the folding and oligomeric state of RihC (Figure 2).

d.     Presentation of kinetic data in Table 3 is difficult to interpret. The presentation of their kinetic data needs to be improved. The authors should consider graphical ways to present their kinetic data, especially in comparison to multiple homologues across multiple substrates. At the very least, the authors need to reorganize the table to better separate the kinetic data, flipping from vertical to horizontal might accomplish this necessary revision.

e.     Figure 4: Kinetic data presented needs error bars to help with the interpretation of differences across substrates and proteins.

3.     At the end of the results/discussion section (pages 19-20: lines 616 – 675), the authors dissect a key loop that shifts positions in the structures of RihC and homologues and argue that changes in this loop positioning are one possible explanation for the shifted substrate specificity and catalytic activity of RihC. The hypothesis represented in this section is reasonable but could use more direct support from figures that more clearly illustrate this loop and the associated changes in RihC structure. Some proposed changes to solidify this explanation are given below:

a.     Add structural close-ups of this region clearly illustrating the changes in this loop position, its relationship to the active site, and its positioning relative to the ligand bound structures.

b.     Take the relevant part of the alignment (SI figure 7) for this region and move it into the main article. Potentially show as a LOGO diagram to more clearly illustrate the relative conservation of these residues.

c.    Lines 619-629: The authors describe a model that was calculated for this loop since it is unresolved in the RihC structure. The authors need to explain how this model was made and validated. A new figure should also be added that shows the alignment between the original and modeled structure and how this loop is positioned in the modeled version. This can be added

Author Response

Reviewer 4 Answers

In this article, Shaposhnikov et al describe the kinetic and structural characterization of a ribonucleotide hydrolase (RihC) from the lactobacilli Limosilactobacillus reuteri. Using a newly developed HPLC based kinetics assay, the authors initially describe the detailed kinetic characterization of RihC against a variety of ribonucleosides and deoxyribonucleosides, finding that RihC requires the 2’OH to catalyze their reactions. Kinetic conclusions are however muddled by complex product inhibition by one product, cytidine, and extrapolation of a single point into larger conclusions about substrate activation with uridine. Conclusions are also made about kinetic differences based on the location of a histag. However, given the distant location of either terminus from active site and the identical KM for the N-terminal and C-terminal histag variants, these shifts in kinetic activity are more likely due to minor variations in enzyme concentrations and do not reflect intrinsic effects of histag location on catalytic activity. The authors really build a more complete story with the determination of the structure of RihC and detailed comparison to similar homologues. Through these structural comparisons, the authors build a compelling story about small structural shifts in dynamic loops that provide differential catalytic activity in RihC versus homologues. With the revisions outlined below and additional documents prepared for reviewers, this article will be prepared for publication in IJMS.

Thank you for reviewing our manuscript!

Lack of necessary material for review:

1.The authors have not provided the necessary access for the reviewers to a preprint of a submitted article on their kinetic assay, which is essential to interpreting their kinetic results. They also did not provide review access to their submitted but not released PDB file so that reviewers can confirm the proper structure and structural variables for their submission.

Thank you for your comment. We apologize for not providing necessary files during the submission of this article. We provide the draft of our (now) accepted article on developing an approach for the kinetic studies of RihC with the acceptance letter as a screenshot as well as .pdb file of our crystal structure with this revision. Since the mentioned paper has recently been published online, we’ve made the respective reference in the text of the manuscript with its details to make it available for the readers (new reference [12]).

Revisions to article:

2.Issues with kinetic results and interpretation

  1. Figure 3B for kinetics with cytidine indicates strong product inhibition in the measured kinetics. The authors should analyze this product inhibition and include this product inhibition in their kinetic analysis.

Thank you for this comment. While this was an interesting effect that we’ve got for both cytidine and uridine (and maybe for other nucleosides were they soluble in such high concentrations in our working buffer) and at first, we decided to share this with other people this was not the primary subject of our studies. Since these effects do not affect our kinetic studies whatsoever, we decided to exclude previous version of Figure 3 from the manuscript entirely. All Michaelis-Menten dependencies for all studied substrates for both enzyme forms that we used to obtain kinetic parameters of said forms are now shown in Supplementary Figure S4.

  1. For fitting kinetic data with product inhibition, see Variot, Cillian, et al. "Mapping roles of active site residues in the acceptor site of the PA3944 Gcn5‐related N‐acetyltransferase enzyme." Protein Science 32.8 (2023): e4725.

  1. Figure 3A for kinetics with uridine. The authors make multiple conclusions about the increase in the rate at the highest concentration of uridine and state that uridine may lead to enzyme activation at high concentrations.

Thank you for this comment. As mentioned in the previous point we decided to exclude this figure from the article since it wasn’t our primary goal of studies and didn’t affect results in any way. This is indeed an interesting subject that should be studied separately.

  1. This conclusion is not supported by the data as this change is only reflected in a single point presented without error values. Additionally, no other points are presented between 50 and 100 mM showing that this trend is linear and shifts over concentration. If the authors want to make this conclusion, they need to conduct additional kinetics between 50-100 mM and present their kinetic values with errors shown to confirm that these shifts are outside the error of the measurements.

While we decided to exclude the velocity-on-concentration in high range of concentrations plots for uridine and cytidine we still would like to clear some things to not cause a misunderstanding. All rates of reaction on all kinetic plots (both the now omitted plots for uridine and cytidine in high range of concentrations and normal plots for obtaining kinetic parameters) are presented as single dot on each plot, however, each of this dot was obtained as an average of three or more separate enzymatic rate measurements and each of these measurements consisted of five nucleobase concentration measurements each measured three separate times for results convergence. In short, one dot on the plot was obtained from 45 independent analyses for convergence purposes and for the sake of simplicity we averaged all points into one for each of the dot on the plot.

  1. Even with these additional measurements, this substrate activation is unlikely as the authors did not present evidence for a potential allosteric site within their structure of RihC that would be required for this substrate activation.

  1. Figure 4 and Table 3. The authors present a large amount of kinetic data showing that the N-terminal histag version of RihC is higher in activity than the C-terminal histag version of RihC. The authors interpret these shifts as representing an intrinsic impact of the histag on the kinetic properties of RihC. However, a simpler explanation that fits more clearly with their data is that the concentrations for these two histag versions are off slightly and this skews the relative data for only the kcat values for C-terminal histag version. The authors need to revise the overarching conclusions about the histag’s impact on measured kinetics.

There are two points we would like to discuss here. First is that like it is mentioned in “Carrying out an enzymatic reaction” section of Experimental RihC (both forms) for consistency’s sake was used in 200 µg mL-1 concentration for all of the kinetic studies. Second is that in Table 3 we give kcat values for all the substrates and both forms that are independent of enzyme’s concentration and are in s-1. Thus, we think it is appropriate to assume that in the case of our LreRihC His-tag does affect turnover rates and HisC variant is slightly worse than HisN variant in that regard.

  1. When looking at the kinetic data in Figure 4, the relative difference between the N-terminal and C-terminal data stays constant and indicates a systematic difference between the concentration of these proteins.

All of the experiments were carried out with the same protein concentrations for both forms for consistency. We think that the difference in kcat is playing the key role here.

  1. Additionally, in Table 3, the KM values for every substrate with the C-terminal and N-terminal histag versions are nearly identical, indicating that the only change in the kinetics is in the Vmax/kcat values, which is dependent on the enzyme concentration, and that the enzyme concentration is a more likely explanation for these observed differences in kcat values.

We agree that Vmax is dependent on enzyme’s concentration. However, we did not present Vmax values in the manuscript and when we calculated kcat using these Vmax values we took into account enzyme’s concentration and so the kcat values given in the article are in s-1 and independent of enzyme’s concentration.

iii. Their own structural analysis (Figure 1) shows that the two termini are located nearby in space and far distant from the active site, indicating that histag placement is unlikely to impact the properties of RihC. Histag placement also has no impact on the folding and oligomeric state of RihC (Figure 2).

While Figure 1 indeed shows that both N- and C-termini are located far from the active site of RihC this figure also shows that while N-tag should be pointed fully away from the protein globule C-tag seems to be pointing a bit towards it. Of course, this is just an assumption since no modeling is perfect as of right now but since we’ve got experimental data (both the expression data and kinetics) backing up the fact that N-tag seems to be better than C-tag for our enzyme we decided to try to explain these phenomena with the difference of His-tag location.

  1. Presentation of kinetic data in Table 3 is difficult to interpret. The presentation of their kinetic data needs to be improved. The authors should consider graphical ways to present their kinetic data, especially in comparison to multiple homologues across multiple substrates. At the very least, the authors need to reorganize the table to better separate the kinetic data, flipping from vertical to horizontal might accomplish this necessary revision.

We thank you for this comment. In the original word file of the manuscript that we attached first when we submitted our manuscript this table was in album orientation and was in our opinion quite fine to read and interpret. However, the newest file that we got for revisions got stylistically changed into the IJMS paper template (even if we ourselves used all the same fonts and everything) and said table was inserted with book orientation making it quite difficult to read and unpleasant to look at. While we don’t know which version of the manuscript you’ve read we still changed the table flipping it to look better in book orientation.

  1. Figure 4: Kinetic data presented needs error bars to help with the interpretation of differences across substrates and proteins.

Thank you for this comment. We’ve made necessary adjustments to the Figure 4 (now Figure 3).

  1. At the end of the results/discussion section (pages 19-20: lines 616 – 675), the authors dissect a key loop that shifts positions in the structures of RihC and homologues and argue that changes in this loop positioning are one possible explanation for the shifted substrate specificity and catalytic activity of RihC. The hypothesis represented in this section is reasonable but could use more direct support from figures that more clearly illustrate this loop and the associated changes in RihC structure. Some proposed changes to solidify this explanation are given below:

We’ve rechecked and recalculated our kinetic parameters and made necessary adjustments to the structural discussion section in the manuscript. The explanation on this is given at the end of this document.

  1. Add structural close-ups of this region clearly illustrating the changes in this loop position, its relationship to the active site, and its positioning relative to the ligand bound structures.

Thank you for this comment. We’ve provided this closed-up figure as a supplementary file for you. The relative positions of the three regions in question are shown in (now) Figure 5E-G.

  1. Take the relevant part of the alignment (SI figure 7) for this region and move it into the main article. Potentially show as a LOGO diagram to more clearly illustrate the relative conservation of these residues.

We thank you for this comment. However, we think it should be fine leaving the alignment in supplementary as a full since we carefully pointed out all the necessary regions in the text. We think that inserting only part of the alignment into the main text might confuse readers and clutter the manuscript.

  1. Lines 619-629: The authors describe a model that was calculated for this loop since it is unresolved in the RihC structure. The authors need to explain how this model was made and validated. A new figure should also be added that shows the alignment between the original and modeled structure and how this loop is positioned in the modeled version. This can be added

The modeling was performed by using two different approaches as described in the article. The first model was for determining the preferrable His-tag position and the second was to resolve the unresolved regions of the crystal structure. The information on obtaining both model structures is presented in “Obtaining model protein structures” section of Materials and methods. The alignment of LreRihC crystal structure and LreRihC model with resolved regions is presented in Figure 9B (now Figure 8B).

We would also like to sincerely apologize to the reviewer for the mistake in our calculations that we haven’t noticed before. When we conducted enzymatic reactions, we first diluted our enzyme 10 times and then used this solution for the reaction itself. However, when we added 5 µL of the enzyme to the 500 µL reaction solution our enzyme was also diluted 100 times additionally. When we calculated kcat values for all of the substrate reactions we forgot to take into account this 100 times dilution of our enzyme and thus our kcat values were 100 times lower than they should have been. We’ve carefully checked and recalculated all of our data for this revision and now present the correct kcat values in our revised manuscript. We’ve also rewritten the appropriate parts of kinetic and structural discussion sections.

Round 2

Reviewer 4 Report

Comments and Suggestions for Authors

I appreciate the multiple edits that the authors have made to this article in response to the original review. 

The only ongoing edit that I have about the article is in regards to the proposed differential kinetic activity between the C-terminal and N-terminal histag versions. The authors misunderstood my original criticism but I still think that my critique holds valid for their data.

I understood that the authors were reporting kcat values, which are semi-independent of enzyme concentration. To calculate these kcat values, the authors took the Vmax value and then divided by the [E]. I also understand that the authors claim that the concentrations were both 200 ug/mL for both of these two variants. However, I am assuming that the authors only purified these proteins once and used a single preparation of enzyme to make all of the kinetic measurements. This means that all of the calculated kcat values are subject to systematic error due to variations in the measured enzyme concentrations, which were used to calculate the kcat values. If the C-terminal variant concentration is actually slightly lower than expected even within error for protein concentration, this would make all of its kcat values appear higher or if the N-terminal variant concentration might be too high, again within error for the protein concentration measurement, this would make its kcat values appear lower. 

If the authors want to make these claims about these differences in the kinetics based on the histag location, then they would need to repeat these kinetics with a separate batch of each protein to confirm that these small kcat differences are reproducible and are independent of a single protein concentration measurement. 

Instead, the authors could chose to remove this extended discussion of these kinetic differences and conclude that the kinetics of the N-terminal and C-terminal variants are the same within the error of the entire experiment, including protein purification and protein concentration measurements. This is especially warranted because the interpretation of these differences is hand-waving at best. 

After this revision is complete, I think that the article is prepared for publication in IJMS.

Author Response

Reviewer 4

I appreciate the multiple edits that the authors have made to this article in response to the original review.

The only ongoing edit that I have about the article is in regards to the proposed differential kinetic activity between the C-terminal and N-terminal histag versions. The authors misunderstood my original criticism but I still think that my critique holds valid for their data.

I understood that the authors were reporting kcat values, which are semi-independent of enzyme concentration. To calculate these kcat values, the authors took the Vmax value and then divided by the [E]. I also understand that the authors claim that the concentrations were both 200 ug/mL for both of these two variants. However, I am assuming that the authors only purified these proteins once and used a single preparation of enzyme to make all of the kinetic measurements. This means that all of the calculated kcat values are subject to systematic error due to variations in the measured enzyme concentrations, which were used to calculate the kcat values. If the C-terminal variant concentration is actually slightly lower than expected even within error for protein concentration, this would make all of its kcat values appear higher or if the N-terminal variant concentration might be too high, again within error for the protein concentration measurement, this would make its kcat values appear lower.

If the authors want to make these claims about these differences in the kinetics based on the histag location, then they would need to repeat these kinetics with a separate batch of each protein to confirm that these small kcat differences are reproducible and are independent of a single protein concentration measurement.

Thank you for your kind words and your comment. We understand that we didn’t make things clear in the manuscript and we apologize for that.

When we studied kinetics we didn’t use only one batch of each enzyme form even though we now see that we didn’t make this clear. First, we studied enzyme’s kinetics with uridine (for both forms) several times with one enzyme batch to see if the kinetics we got are within the margin of error. Both KM and kcat for both forms (even though we initially forgot to take into account one of the dilutions and got wrong kcat values) stayed roughly the same within the error. Then for each consecutive substrate that we studied kinetics for we’ve purified a new batch of enzymes (both forms) AND we also re-did uridine studies each time to make sure that we’ve got right values. Then we also re-did studies for that substrate one more time to see if kinetic values would be the same. In total, for each substrate we’ve studied kinetics twice (each substrate with a different batch of freshly purified enzyme) except for uridine for which we’ve studied kinetics twice initially and then once each time parallel with the new substrate. All of the values were almost the same (within the error margin), thus we concluded that we studied all the kinetics correctly and it was indeed the position of His-tag that explained the differences in kinetics.

We will add this explanation to the Experimental section of the manuscript.

Instead, the authors could chose to remove this extended discussion of these kinetic differences and conclude that the kinetics of the N-terminal and C-terminal variants are the same within the error of the entire experiment, including protein purification and protein concentration measurements. This is especially warranted because the interpretation of these differences is hand-waving at best.

After this revision is complete, I think that the article is prepared for publication in IJMS.
